# System Redesign: The Value of a Primary Care Liaison Model to Address Unmet Social Needs among Older Primary Care Patients

**DOI:** 10.3390/ijerph182111135

**Published:** 2021-10-23

**Authors:** Jungyoon Kim, Valerie Pacino, Hongmei Wang, April Recher, Isha Jain, Vaibhavi Mone, Jihyun Ma, Mary Jo Spurgin, Daniel Jeffrey, Stephen Mohring, Jane Potter

**Affiliations:** 1Department of Health Service Research & Administration, University of Nebraska Medical Center, 984350 Nebraska Medical Center, Omaha, NE 68198, USA; valerie.pacino@unmc.edu (V.P.); hongmeiwang@unmc.edu (H.W.); isha.jain@unmc.edu (I.J.); vaibhavi.mone@unmc.edu (V.M.); 2Nebraska Medicine Nebraska Medical Center, Omaha, NE 68198, USA; arecher@nebraskamed.com; 3Department of Biostatistics, University of Nebraska Medical Center, Omaha, NE 68198, USA; jihyun.ma@unmc.edu; 4Department of Internal Medicine, Division of Geriatrics, Gerontology and Palliative Medicine, University of Nebraska Medical Center, Omaha, NE 68198, USA; maryjo.spurgin@unmc.edu (M.J.S.); jpotter@unmc.edu (J.P.); 5Department of Internal Medicine, Division of General Internal Medicine, University of Nebraska Medical Center, Omaha, NE 68198, USA; djeffrey@unmc.edu (D.J.); smohring@unmc.edu (S.M.)

**Keywords:** older adult, health promotion, what matters, primary care liaison, social determinants of health

## Abstract

Assessing and addressing social determinants of health can improve health outcomes of older adults. The Nebraska Geriatrics Workforce Enhancement Program implemented a primary care liaison (PCL) model of care, including training primary care staff to assess and address unmet social needs, patient counseling to identify unmet needs, and mapping referral services through cross-sectoral partnerships. A PCL worked with three patient-centered medical homes (PCMHs) that are part of a large integrative health system. A mixed-methods approach using a post-training survey and a patient tracking tool, was used to understand the reach, adoption, and implementation of the PCL model. From June 2020 to May 2021, the PCL trained 61 primary care staff to assess and address unmet social needs of older patients. A total of 327 patients, aged 65 years and older and within 3–5 days of acute-care hospital discharges, were counseled by the PCL. For patients with unmet needs, support services were arranged through community agencies: transportation (37%), in-home care (33%), food (16%), caregiver support (2%), legal (16%), and other (16%). Our preliminary results suggest that the PCL model is feasible and implementable within PCMH settings to address unmet social needs of older patients to improve their health outcomes.

## 1. Introduction

It is estimated that 40–90% of health outcomes are attributable to social determinants of health (SDoH), social, behavioral, and economic factors that typically fall outside the purview of traditional medical care systems [1]. Identifying and addressing SDoH, such as access to housing, transportation, food, in-home care, and social connection, plays a significant role in improving the health of older adults and potentially reducing health care costs by preventing avoidable hospitalizations.

For older adults, such SDoH significantly influence health, morbidity, mortality, and their ability to live independently and age in place. An American Association of Retired Persons (AARP) Foundation survey of older adults found that more than one-third (34.4%) lived in low-income households and one in five (22.8%) found it somewhat or very difficult to pay their monthly expenses. Food insecurity was reported among 16.2% of respondents, while two in five (40%) respondents were concerned that they would not be able to afford to stay in their homes or make repairs to their homes as they age. With these findings, AARP recommended that SDoH screening and referrals occur at every wellness visit so that health professionals and health systems can take these needs into account in the provision of their health services and make referrals to community partners [2].

The Older American Act (OAA) was signed into law over 50 years ago to provide funding to states to ensure that older Americans could receive community-based social and health-related services (e.g., Meals on Wheels, caregiver support, etc.) through local Area of Agencies on Aging (AAAs) and more recently through Aging and Disability Resources Centers (ADRCs). Yet, despite the dramatic increase of older populations and Medicare expenditures in the U.S. over the last decades, funding for the OAA has been flat, limiting the proposed reach of the OAA [3]. In 2017, the Centers for Medicare and Medicaid Services launched Accountable Health Communities (AHCs) to address SDoH at 28 organizations to explore how building support within communities might improve health care outcomes [4]. While this model could be a longer-term solution, it is imperative that local solutions be sought that can be put into place more immediately.

Linking primary care with community organizations and resources holds great promise, but only if resources are available, accessible, affordable, and perceived as valuable [5,6]. To these ends, cross-sectoral partnerships between health care, social services, and other community organizations appear to be critical to help older populations meet social and medical needs and have demonstrated success in improving function and quality of life, decreasing health care utilization and spending (e.g., 30-day readmission rates), and reducing preventable deaths [7,8,9,10]. Brewster et al. found that counties whose AAAs maintained informal partnerships with a broad range of organizations in health care and other sectors had lower hospital readmission rates, making the agencies a natural point of intervention for efforts to foster effective cross-sectoral partnerships to serve the complex needs of geriatric populations [11].

A value-based healthcare delivery model enables innovations in care management that serve geriatric populations. A safety-net accountable care organization (ACO) in Minnesota has hired community health navigators to connect Medicare beneficiaries with vocational services and affordable housing opportunities. This resulted in nearly 10% reductions in emergency department visits, and increased outpatient visits and savings [12]. Among all the value-based organizations, patient-centered medical homes (PCMHs) appear particularly promising [13,14]. The PCMH model emphasizes multidisciplinary team-oriented care, a holistic approach, and coordinated care, which entails and manages patient care across various health systems and community services, with an emphasis on efficient and safe transitions of care [15].

The Age-Friendly Health Systems movement, initiated in 2017 by the John A. Hartford Foundation, recognizes the need for embracing all sectors, from primary care to community-based organizations, to provide seamless care and services to older adults that respect their goals and preferences, guided by the evidence-based framework “4Ms”—What Matters, Mentation, Mobility, and Medication [16]. What matters entails knowledge about the goals and care preferences of each older adult and aligning the care delivery accordingly. Mentation includes preventing, identifying, treating, and managing dementia, delirium, and depression across settings of care. Mobility encompasses ensuring safe movement every day to continue function and execute what matters for each older adult. Medication involves only using when necessary and prescribing age-friendly medication that does not impede what matters, mentation, and mobility in older adults [17]. Our PCL model is based on the premise of what matters to the majority of older adults, who want to remain at home as independently as possible.

The Nebraska Geriatrics Workforce Enhancement Program (NGWEP) adapted a primary care liaison (PCL) model from the Northwest Geriatrics Workforce Enhancement in King County, Washington [18]. Our PCL intervention integrates many of the findings of previous research, including (1) robust training of primary care staff in how to assess and address unmet social needs, (2) mapping referral services and implementing other models of care when unmet needs are identified, and (3) cross-sectoral partnerships between clinic partners, and the Eastern Nebraska Office on Aging, the Area Agency on Aging that serves the five-county region around Omaha, Nebraska.

We hypothesized that the PCL model is implementable and feasible and is particularly well-suited for PCMH settings. The goal of the study was to describe the development of the PCL model and evaluate the early phase of PCL implementation to answer the following questions: (1) Can the PCL model increase awareness of PC providers/staff on SDoH by delivering education? (2) Can the PCL model be implemented in PCMH clinics efficiently and identify specific patients where nonmedical needs are impairing medical care? (3) Can these patients be referred to community partners, so their nonmedical needs can be adequately addressed?

## 2. Materials and Methods

### 2.1. Design of the Study

To answer our research questions, we designed a descriptive study using a mixed-methods approach that combines quantitative and qualitative data to glean a comprehensive understanding of the reach, adoption, and implementation of the PCL program. The University of Nebraska Medical Center Institutional Review Board determined that this project is not considered human subjects research because it is focused on organizational quality improvement (IRB waiver # 651-19-EP).

### 2.2. Setting

The study took place in PCMH clinics as a part of a large integrative health system in a mid-Western city in the U.S. Three out of the 15 PCMH clinics in the system were selected for the early phase of PCL implementation. Clinic A (pilot) was selected because a large portion of patients were older, and the NGWEP project director serves as medical director at this clinic. Clinic B and C were chosen because they both serve low-income, racial/ethnic minority populations in the network. All PCMH clinics in this system have interprofessional teams that include medicine, nursing, advanced practice providers, social workers, behavioral health therapists, nutritionists, and pharmacists. Nursing staff is organized as: RN nurse care coordinators (focused on care between outpatient visits), LPNs, and medical assistants who participate in outpatient visits, monitor medication refills, and referrals.

### 2.3. Target Population 

Patients aged 65 years and older with a primary care provider, who were at one of the three clinics, and who were discharged from the hospital within 3–5 days, were eligible for PCL intervention. The PCL used the electronic medical record (EMR) system to identify newly discharged patients from the participating clinics.

### 2.4. Intervention

As illustrated in Figure 1, the PCL was hired by the same health system as the PCMH clinics. As a health system employee, the PCL has a full access to the health system’s EMR and email systems. The PCL has an office space at ENOA within the Information and Referral Department and has access to the ADRC and other state information systems. With this unique position, the PCL is able to connect patients to relevant departments in community organizations for specific needs. Funding support for the PCL is split between the NGWEP and the health system in year 1, with the health system assuming an increasing portion of the cost for years 2–5. 

The functions of the PCL include:Increasing awareness of SDoH and community resources among PCMH providers/staff by educating them on how to identify and address patients’ social needs through the PCL program;Reaching out to recently discharged geriatric patients by phone (one to three times) and assessing patients’ unmet social needs as part of the PCMH transitional care team; andConnecting patients to community services provided by ENOA and other agencies.

Between August 2020 and April 2021, the PCL reached out to the three PCMH practices to educate their providers/staff on SDoH and community resources. The trainings were generally 15–20 min long, delivered via a livestreaming online conference tool (Zoom) during the clinic’s staff meeting.

The participating PCMH clinics worked to improve patients’ transitions of care from hospital to home over the 2 years prior to the PCL intervention. The prior transition of care intervention included telephone calls by a nurse-care coordinator (NCC) to assess medical needs within 48 hours of hospital discharge. The NCC call focuses on medications changes during the hospital stay, access to new medications, recurring and new symptoms, need/access to home-health care, equipment needs, and arrangements (transportation to) for the follow-up appointment 1–2 weeks after discharge. The PCL intervention supplements the medical transition with calls placed within 48–72 hours of discharge to assess adequacy of the discharge location, financial problems (whether related to the hospital stay or not), food in the home and meal preparation, problems completing activities of daily living, and availability of someone to help if needed.

### 2.5. Evaluation Frameworks

This study used the Reach, Effectiveness, Adoption, Implementation, and Maintenance (RE-AIM) framework to evaluate the feasibility and acceptability of the model in identifying and connecting older patients to community services [19]. The RE-AIM framework is well-established in evaluation and implementation science and allows us to evaluate the program on individual and organizational levels. Table 1 describes the three RE-AIM dimensions and how the PCL intervention was assessed for each dimension. 

### 2.6. Data and Measurements

Data for the Reach, Adoption, and Implementation measures were derived from a PCL patient tracking log (an Excel spreadsheet maintained by the PCL to organize and track outreach efforts). The log contains the date and number of PCL attempts to eligible patients, whether the PCL screened patients’ social needs, type of recommended services, follow-up information of the service referral, and qualitative notes on reasons for no service referrals. We used a web-based survey (REDCap system) to assess improvements in PC providers’ knowledge of and confidence in identifying and addressing SDoH. We used a single item for knowledge and confidence on a 5-point Likert scale (1 = Very Low to 5 = Very High) using retrospective pre-post questionnaires. 

Reach was calculated by counting the number of eligible patient cases contacted by the PCL and had unmet social needs assessed (those exposed to the intervention), and its proportion of all geriatric patient cases with recent hospitalizations (those who were eligible for the intervention). Clinic-level adoption was determined as “adopted” if a single patient case was successfully contacted, evaluated, and referred to community resources by the PCL. Provider-level adoption was assessed by comparing median scores of staff knowledge and confidence before and after the SDoH training. Implementation was calculated by counting the number of patient cases who were referred to community services through the PCL intervention, and its proportion of all patient cases whose needs were identified. Reasons for no referral were examined using standard qualitative content analysis [20].

### 2.7. Analysis

Descriptive statistics (means, standard deviations, median, interquartile range, frequencies, and percentages) were used to characterize reach, adoption, and implementation. We used the Wilcoxon-ranked test to assess changes in PC providers’ and staff’s knowledge and confidence before and after the training led by the PCL at a level of significance (alpha = 0.05). All quantitative data analysis was performed using SAS version 9.4 (SAS Institute Inc., NC, USA).

## 3. Results

### 3.1. Clinic Characteristics

Clinic A is a geriatrics teaching clinic located in the central city a few blocks from clinic C. All patients of clinic A are insured by Medicare/Medicare Advantage (federal health insurance program for people 65 years old and older) and many are dual-eligible for Medicaid. About 80% of patient visits are for primary care and 20% for consultation. The mean age of patients is about 78 years old and about 65% are women. The clinic is staffed by 10 academic geriatricians, 2–3 geriatric medicine fellows, and 3 Advanced Practice nurses. 

Clinic B is located in an area of the city occupied by a black/African American and immigrant population. Clinic B has established strong ties to the black/African American community through decades of services to the community. About two thirds of clinic B’s patients are non-white minorities (59% African American, 3% Hispanic, 3% Asian). Compared to the entire city’s proportion of minority populations (8.1% African American, 12.3% Hispanic, and 3.4% Asian), clinic B serves much higher proportions of African American patients. Insurance coverage for the clinic population is 31% Medicaid, 27% Medicare or Medicare Advantage, and 9% self-pay. Based on the zip codes for clinics B’s service area, 25.6% live in poverty (city average = 13.4%).

Clinic C is located in the central city and serves a lower-income urban population. This is a resident-only clinic. While there are 18 MD preceptors, only the 68 medical residents see patients at the clinic C for one day every two weeks. Of the 3150 unique patients, 19% of patients are uninsured and, based on zip codes for the service area, 28% live in poverty, which is higher than the entire city’s poverty rate of 13.4% [21]. Medicare insures 40% of patients, 17% receive Medicaid, and 25% are 65 years old or older. More than 60% of patients are non-Hispanic white and 24% are African American.

### 3.2. Reach

During the initial phase of PCL intervention (5 June 2020–20 April 2021), the PCL attempted to contact a total of 406 eligible patient cases within 3–5 days of discharge from acute-care hospitalization. Among those, the PCL reached 327 (80.5%) patient cases by phone to discuss their needs for social and support services, and 323 patients (79.6%) were screened for their needs for community services, such as transportations, food services, in-home care, and more.

### 3.3. Adoption

#### 3.3.1. Clinic-Level Adoption

The PCL services were adopted 100% for all three PCMH clinics by successfully contacting, screening for nonmedical needs, and connecting at least one patient at each clinic to community services during the initial phase of implementation. 

#### 3.3.2. Provider-Level Adoption

Raising staff awareness by education is one of the primary functions of the PCL. If PC providers/staff clearly understand the need for the program and are able to identify unmet social needs in their patients, the PCL program may be more successfully adopted in practice. Between June 2020 and April 2021, the PCL educated a total of 61 interprofessional PC providers/staff on how to identify and refer patients to address unmet SDoH. Table 2 shows the breakdown of attendees by professions and disciplines.

Our post-training evaluation survey showed improved staff knowledge and confidence in identifying and addressing social determinants of health for older patients (Table 3). On a scale of 1 (very low) to 5 (very high), trainees reported significant improvement in knowledge (median diff = 1.0; *p* = 0.0039) and confidence (median diff = 1.0; *p* = 0.0156).

### 3.4. Implementation

As illustrated in Table 4, of the 323 screened, 43 (13.3%) had unmet needs, and were referred to community services through ENOA and other agencies. The most frequently needed service was transportation (37.2%), followed by in-home services (32.6%), food (16.3%), caregiver support (2.3%), legal services (16.3%), and other (16.3%). 

We conducted an additional content analysis for those 280 patient cases whose needs were screened, but no referral was made. The most frequent reason why referral was not made was because patients or caregivers identified no needs for social and community services, thus denied services (*n* = 230). Other reasons included patient had services already (*n* = 41), patient refused to cooperate with the questions (*n* = 4), patient wanted services but were unable to receive them at this time (*n* = 2), and patient was currently staying in a facility (*n* = 3). 

## 4. Discussion

Social determinants are essential to health outcomes, yet health systems and the community-based service providers best positioned to meet social needs are traditionally separate. As population health grows in importance within health care systems, bringing the two together deserves attention [22]. Here, we describe one such effort that creates a position (a primary care liaison) within a health care system whose role is to link the health care system to community-based services. As illustrated in Table 5, the creation of the PCL position brought changes in the patient referral processes to address SDoH before and after implementing the program. Hired by the health system, the PCL has real-time access to EMR to identify eligible patients after discharge. It also complemented social workers’ roles by enabling social needs assessment and ensuring that patients are connected to community services after discharge. Our system redesign approach resulted in more than 300 older patients being educated and screened for their nonmedical needs after discharge, and 43 actual services arranged by the PCL.

This study sought to determine whether the PCL model implemented in primary care settings can efficiently identify specific patients where nonmedical needs are impairing medical care, refer patients to community partners who will address those nonmedical needs, and in doing so, improve their medical outcomes. Our preliminary findings from the early phase of PCL implementation showed that the model is acceptable, implementable, and effective at addressing the unmet needs of social determinants of health of older adults. These findings are consistent with those of reviews of PCL models, including the one on which our approach was modeled. Boll et al. found that the PCL model is feasible, improves interactions between primary care providers and AAAs, and connects older adults and their caregivers to resources to address their unmet SDoH [23].

Of the 323 patient cases educated and screened for SDoH, only 43 patient cases were referred to community services and 280 were not. We compared the demographic characteristics of these two groups of patients. We found no meaningful differences in age, gender, and ethnicity between the two groups. For race, the referral group has a higher portion of non-white minority than that of the non-referral group. This is consistent with our findings from the content analysis that the non-referral group has fewer identified needs on social determinants of health. 

The study findings suggest that a PCMH setting is a suitable place to start building local-level cross-sectoral collaborations to address unmet social needs of older patients through the PCL model. With their approach to comprehensive, coordinated, patient-centered care that is accessible and safe, PCMHs may be uniquely appropriate for managing the complex biopsychosocial needs of older patients. Given their focus on population health outcomes, PCMHs have incentives to focus on the nonmedical aspects of geriatric care.

It should be noted that healthcare professionals and managers’ participation in developing the intervention allowed us to better tailor the intervention to the target setting and target population. We closely communicated with PCMH leaders, each clinic’s Geriatrics Champion, practice managers, and community organization representatives throughout the intervention development and implementation process. Additionally, communication with clinic staff on the PCL role facilitated the implementation and quality of PCL referrals. One challenge during the initial phase of the program was that sometimes there was unclear understandings of the roles and responsibilities between this new role (PCL) and existing social worker roles. Emphasizing the bridging role of PCL for community resources and effective coordination will help sustain the PCL program within PCMH settings. 

With an eye toward sustainability, the PCL position transitioned from partial support by the health system to full support by the end of the project. This agreement was formulated during planning sessions with key clinical and administrative leaders at the health system and PCMH network. Negotiations on this relationship occurred over three months before the program started and involved face-to-face meetings between leadership from both the health system and ENOA. The Medical Director of the PCMH network collaborated in this process, and using existing data tracking on care transitions, ER visits, and readmissions, the health system can measure the cost-effectiveness of this investment on behalf of NGWEP and our clinic partners.

### Limitations

Although this report shows that a PCL model is feasible and implementable within PCMH clinics, justifying such a model for support by a health care system will need to demonstrate improvement in health outcomes for the population served by the intervention. Others [7,8,9,10] report that partnerships between health care, social services, and community organizations help older populations meet both social and medical needs and show improvement in function, quality of life, and decreased health care costs, including reduced 30-day readmission rates. A planned second phase of this study will examine health outcomes. Future study may also consider examining patient experience or satisfaction with the PCL services.

Our findings were based on the three PCMH clinics within a large integrated health system located in a Midwestern city. The results may not be generalized to different types of PCMH or primary care settings. The geriatric population served by these three PCMH clinics were not representative of the entire older population in the city. Different system-level factors, such as leadership, power relations, affiliation type, and IT infrastructure, as well as patient-level factors, such as primary languages and culture of the target population served, should be considered for successful implementation of the PCL model. 

The study has limited implementation fidelity information, such as program differentiation, adherence, exposure, quality, and responsiveness. A comprehensive assessment of implementation fidelity will allow us to make stronger inferences about effectiveness outcomes. Most meaningfully, if we do not find improved patient outcomes and high implementation fidelity, we can conclude that our intervention did not work as intended. If we find improved patient outcomes but low implementation fidelity, we may conclude that there were external factors at play. However, if we find improved patient outcomes and high fidelity, our conclusions about efficacy will be stronger.

## 5. Conclusions

In this quality improvement project, we report data describing how a partnership between an Area agency on Aging (ENOA) and a health care system can address social determinants of health at the time older people are discharged from the hospital. In this model, the Primary Care Liaison (PCL) is an employee of and paid by the health care system. As a health system employee, the PCL has full access to the electronic medical record (EMR) to identify discharged patients and document work on social determinants of health within the EMR. That documentation is visible to the system’s health care providers. The area agency provides office space and access to the agency’s exhaustive information system on community-based services. The area agency’s Information and Referral Department’s personnel also collaborate with the PCL to connect some clients with service needs. 

We also report our early findings in PCL implementation. During the initial phase of implementation from June 2020 to April 2021, the PCL reached 327 older patients by phone to discuss their unmet social needs. Of those whose unmet needs were identified (*n* = 323), the PCL connected 43 patients with community support services, including in-home services, transportation, food, caregiver support, and other. Our study shows promise for a PCL model to identify and address unmet social needs of older patients of PCMH clinics by linking them with community-based services after hospital discharge. Future work on the PCL model will include a cost-effectiveness analysis to explore cost savings to the health system that warrant ongoing support for PCL services.

## Figures and Tables

**Figure 1 ijerph-18-11135-f001:**
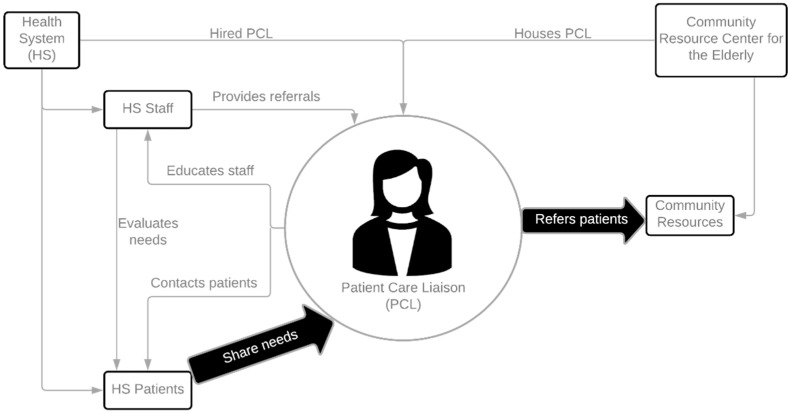
System redesign feature of the primary care liaison model.

**Table 1 ijerph-18-11135-t001:** RE-AIM framework applied to the evaluation of the PCL intervention.

RE-AIM Dimension	Primary Focus	PCL Evaluation Measure
Reach	How do I reach the targeted population with the PCL intervention?	Number and proportions of eligible patient cases contacted and screened by the PCL for their nonmedical needs
Adoption	What are the clinic/system level support to deliver PCL intervention effectively?	At least one patient case per clinic contacted and referred to community resources by the PCL PC providers/staff knowledge/confidence improved on SDoH and community resources
Implementation	How do I ensure that the PCL intervention refers patients to receive needed community services?	Number and proportions of patient cases referred to community services, or services arranged by the PCL Reason for no referral cases

**Table 2 ijerph-18-11135-t002:** Primary care staff education on social determinants of health by the PCL.

	Number of PC Providers/Staff by Professions and Disciplines					
	Behavioral Health ^1^	Medicine ^2^	Nursing ^3^	Physician Assistant	Other ^4^	Total
Session 1	2	12	5	1	5	25
Session 2	0	0	5	0	0	5
Session 3	1	12	9	2	7	31
All	3	24	19	3	12	61

^1^ Behavioral health includes clinical social work. ^2^ Medicine includes family medicine, internal medicine, geriatrics, and psychiatry. ^3^ Nursing includes RN, NP, LPN/LVN, and other nurses. ^4^ Other includes administrator, pharmacist, medical assistant, student, and profession not listed.

**Table 3 ijerph-18-11135-t003:** Change in primary care staff knowledge and confidence ^1^.

Topics	N	Before		After		Difference		*p*-Value
		Mean (SD)	Median [IQR]	Mean (SD)	Median [IQR]	Mean (SD)	Median [IQR]	
My knowledge of the SDoH for older adults	10	2.7 (0.48)	3.00	3.6 (0.52)	4.00	0.9 (0.32)	1.00	0.0039
			[2–3]		[3–4]		[1–1]	
My confidence in identifying and addressing SDoH for older patients	10	2.8 (0.63)	3.00	3.5 (0.53)	3.5	0.7 (0.48)	1.00	0.0156
			[2–3]		[3–4]		[0–1]	

^1^ Scale 1 = Very Low, 2 = Low, 3 = Medium, 4 = High, 5 = Very High; SDoH = Social Determinants of Health; SD = Standard Deviation; IQR = Inter Quartile Range.

**Table 4 ijerph-18-11135-t004:** Number of patients 65 and older screened and referred to community services by primary care liaison.

		Clinic A	Clinic B	Clinic C	All
Patients eligible ^1^		205	74	127	406
Patients screened for unmet needs, *n*		170	59	94	323
Patients referred to community services, *n* (%)		21 (12.4)	14 (23.7)	17 (18.1)	43 (13.3)
Type of services Arranged ^2^	Transportation services, *n* (%)	7 (38.9)	6 (54.5)	3 (21.4)	16 (37.2)
	In-home services, *n* (%)	8 (44.4)	3 (27.3)	3 (21.4)	14 (32.6)
	Food services, *n* (%)	3 (16.7)	2 (18.2)	2 (14.3)	7 (16.3)
	Caregiver resources, *n* (%)	1 (5.6)	0 (0)	0 (0)	1 (2.3)
	Legal services, *n* (%)	0 (0)	2 (18.2)	5 (35.7)	7 (16.3)
	Other, *n* (%)	2 (11.1)	1 (9.1)	4 (28.6)	7 (16.3)

^1^ Does not necessarily reflect a unique number of patients, since patients could have multiple hospitalizations during the study period. Each discharge is treated as a separate encounter, after which the PCL attempted outreach. Multiple attempts to contact the same patient after a given discharge were counted only once. PCL service interval: 5 June 2020–20 April 2021. ^2^ Percentages may not add up to 100% due to rounding.

**Table 5 ijerph-18-11135-t005:** Changes in referral process before and after implementation.

Process	Pre-Implementation	Post-Implementation
Methods to identify patients’ SDoH after discharge	Not available	EMR review by PCL and interview of patient/caregiver
Primary person addressing patients’ SDoH	Social workers serve patients in hospital and during clinic visits as needed and as they are available, but not post-discharge as new needs arise	PCL assesses social needs arising after discharge and before patients are seen back in clinic.
Process to connect patients to social services	Inpatient social work attempts to anticipate needs after discharge and make recommendations	PCL ensures that patients are connected to the services requested at discharge and for evolving needs after returning home.

## Data Availability

The data presented in this study are available on request from the corresponding author. The data are not publicly available due to privacy restrictions of study participants.

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
