# Peer review of "System Redesign: The Value of a Primary Care Liaison Model to Address Unmet Social Needs among Older Primary Care Patients"

_ijerph, 2021, doi:10.3390/ijerph182111135_

Round 1

Reviewer 1 Report

The work of Kim et al., explore a pilot study of older adults in order to implement a primary care liaison model.

The most important revision should be adding more results. Some tables are unclear and there are very few results that could effectively be answered the questions reported in the aim.

Other major revisions:

From line 99 to line 108 is the method and the explanation of PCL. Please move into the method section.

What do you mean by "quantitative surveys" please add more detail.

How were the 3 centers were selected with respect to the area? More precisely these 3 centers covered the entire area in order that they are really representative of the population?

Be careful you have to control the number of the tables, there are some mistakes!

Table 1 is not clear. Why are there these differences? Why different types and duration? Please justify and report it.

From line 138 all these are results also if they are a description of the population and characteristics of centers (mean age and % of females., and so on), it could be better to report it into the results section in a sub-paragraph with the related characteristics. You may leave here only general characteristics of the center.

In the results section the paragraph from line 207 to line 212 is not completely clear, please revised and add at the end of the sentence (Table3). 

Table 2: Why did you report min-max and not the interquartile range?

Which are the characteristics of people not reached? How do they differ from the others enrolled?

Table 3 is unnecessary and redundant you already report all the results in the text. 

The discussion should be reviewed: it is necessary to compare and give some more details with what is already known and done in other studies.

Minor revisions:

In line 161 there is "PCP," as an acronym, not the explanation please add it.

In line 279 please control "(cite refs 20-24)"

Reviewer 2 Report

The article touches on a significant problem - taking into account the social needs of the elderly in the provision of medical services. Unfortunately, a review of the literature, which analyzes this problem from a management perspective, is not presented. 

Although the objectives of the study include determining the potential of the PCL model to meet the social needs of elderly patients, the article does not provide data on whether the social needs voiced by patients were met. It seems that it is incorrect to assess the degree of satisfaction of social needs of health care consumers based on only one indicator (the number of services provided).

In section 1, racial characteristics are presented only for clinics B and C. I believe it would have been more revealing to indicate in this section the variance by patient income rather than patient origin. Especially since the study results do not present a racial component. The results presented in Tables 1 and 2 relate to trained medical staff, which does not allow conclusions to be drawn about the problem posed in the title of the article, as it does not indicate patient satisfaction in any way.

Round 2

Reviewer 1 Report

Dear authors,

thanks for your prompt reply and the work that you have made for your manuscript. Now it is more clear and well structured.

Minor revision

1) Please change and review the subparagraphs 2.6 and 2.7:

-move lines from 217 to 224 in 2.6 because refers to data

-add in 2.7 line 216 which index will be reported in the tables (means...)

-add in 2.7 the value of significance

In line 401 since you used all the explanation and not only the acronym please add it also for "EMR".

Author Response

Response to Reviewer 1:

Thank you for your quick review of the revised manuscript and additional minor revision suggestions. We have corrected all the points you have raised in the section 2.6 and 2.7.

- moved lines from 217 to 224 in 2.6 analysis section to 2.7 data and measurement section.

- added "means, standard deviations, medians, interquartile ranges, frequencies, and percentages" in 2.7 data analysis section.

- added a "value of significance at alpha=0.05" in 2.7 section.

- added "electronic medical records (EMR)" In line 401. 

Reviewer 2 Report

The improvements sufficiently clarify the main goal of article. 

Author Response

Response to Reviewer 2:

Thank you for your quick review of the revised manuscript. We sincerely appreciate your time and expertise on this review. We are glad that you found that improvements we made in the revised article sufficiently clarify the main goal of the article.